# The Diagnostic, Prognostic and Therapeutic Role of miRNAs in Adrenocortical Carcinoma: A Systematic Review

**DOI:** 10.3390/biomedicines9111501

**Published:** 2021-10-20

**Authors:** Chrysoula Mytareli, Danae A. Delivanis, Fani Athanassouli, Vassiliki Kalotychou, Marina Mantzourani, Eva Kassi, Anna Angelousi

**Affiliations:** 1Department of Internal Medicine, Laikon General Hospital, Medical School, National and Kapodistrian University of Athens, 11527 Athens, Greece; fani.athanasouli@yahoo.gr (F.A.); vkalotyc@med.uoa.gr (V.K.); mantzourani@gmail.com (M.M.); a.angelousi@gmail.com (A.A.); 2Division of Endocrinology, Diabetes, Metabolism, and Nutrition, Mayo Clinic, Rochester, MN 55905, USA; ddelivanis@gmail.com; 3Department of Biological Chemistry, Medical School, National and Kapodistrian University of Athens, 11527 Athens, Greece; evakassis@gmail.com; 4Department of Propaedeutic and Internal Medicine, Laikon General Hospital, National and Kapodistrian University of Athens, 11527 Athens, Greece

**Keywords:** microRNAs, adrenocortical carcinoma, biomarkers, diagnosis, prognosis, therapy

## Abstract

Adrenocortical carcinoma (ACC) is a rare endocrine malignancy with a dismal prognosis and a high rate of recurrence and mortality. Therapeutic options are limited. In some cases, the distinction of ACCs from benign adrenal neoplasms with the existing widely available pathological and histopathological tools is difficult. Thus, new biomarkers have been tested. We conducted a review of the recent literature on the advances of the diagnostic, prognostic and therapeutic role of miRNAs on ACC patients. More than 10 miRNAs validated by multiple studies were found to present a diagnostic and prognostic role for ACC patients, from which miR-483-5p and miR-195 were the most frequently met biomarkers. In particular, upregulation of miR-483-5p and downregulation of miR-195 were the most commonly validated molecular alterations. Unfortunately, data on the therapeutic role of miRNA are still scarce and limited mainly at the experimental level. Thus, the role of miRNA regulation in ACC remains an area of active research.

## 1. Introduction

Adrenocortical tumors are common and are detected in 5–7% of the general population [1] and up to 10% in the elderly [2]. Adrenocortical carcinoma (ACC) is an uncommon endocrine malignancy with an annual incidence of 1–2 cases per million [3] and with an extremely dismal prognosis with a 5-year survival rate of less than 35% [4]. Currently, the only curative therapy for localized ACC is surgery, although local recurrence is common, ranging from 19 to 34% [5]. Adjuvant treatments, including chemotherapy and radiotherapy, have shown limited therapeutic effectiveness [6]. The most widely used classification system (tumor, lymph node and metastasis (TNM)) seems to be inadequate for predicting patient outcome and survival [7].

MicroRNAs (miRNAs) are small noncoding RNAs of 21–25 nucleotides that regulate genes expression in a sequence-specific manner, inhibiting their expression by targeting the 3′-untranslated region (3′-UTR) of target messenger RNA (mRNA) [8]. MiRNAs are considered epigenetic regulators, involved mainly in the pοst-transcriptional regulation of gene expression [8], and they are found not only in tissues but also in body fluids [9]. More than 50% of protein-coding human genes are predicted to be modulated by miRNAs. Deregulation of miRNAs has been implicated in the pathogenesis of many human diseases, particularly cancer. Τhe link between miRNAs and cancer was brought about by the seminal observation of Croce’s group, who reported that miR-15 and miR-16, two miRNAs located in chromosome 13 (13q14), are frequently deleted in chronic lymphocytic leukemia and function as tumor suppressors [10]. Since then, miRNAs have been studied more intensively in the field of cancer, and growing evidence suggests that altered miRNA expression is involved in the pathogenesis of various types of cancers.

Recent studies have identified miRNAs that have a functional role in adrenal tumorigenesis, including benign and malignant adrenocortical tumors and pheochromocytomas [11]. The role of miRNA deregulation in ACC was first suggested in 2007, when it was discovered that a long noncoding RNA H19 gene transcript [12] was detected in the 11p15 locus, where IGF2 is also located and associated with Beckwith–Wiedemann syndrome, which leads to the development of pediatric ACC [13]. Since then, a number of studies have been performed comparing miRNA expression in ACCs with normal adrenal cortex and adrenocortical adenomas (ACAs) [14]. Given the biological heterogeneity of ACCs and the limitations of the currently used treatments, a better understanding of miRNAs function may serve as a diagnostic, prognostic and potentially therapeutic tool in the management of these patients. In this systematic review, we present a critical summary of the recent observations describing miRNA dysfunction, focusing on their prognostic role in ACCs. 

## 2. Methods

This systematic review was carried out according to the Preferred Reporting Items for Reviews and Meta-Analyses (PRISMA) statement. 

### 2.1. Data Sources and Search Strategy 

To identify studies and determine their eligibility, a systematic search was conducted in the PubMed and Cochrane Databases from 1 April to 15 April, 2021. The references of review articles and of included original publications were also screened for potentially relevant studies. Search terms included the following: “miRNAs”, “adrenal tumours”, “adrenal neoplasms”, “adrenocortical carcinoma”, “molecular biomarkers” and “epigenetics”. The above keywords were also combined with the Boolean operators AND and OR.

### 2.2. Eligibility Criteria for Articles of Inclusion

A total of 1033 articles were retrieved from the search of the databases. After removing duplicates and non-English literature, 887 articles remained. Two of the authors (C.M and A.A) independently examined all potentially eligible titles and abstracts, from which 735 articles were excluded to identify 152 articles of interest. Studies on children (we included only 66 adults >19 years of age), as well as studies including other adrenal diseases or neoplasms (pheochromocytoma, benign adrenal neoplasms or hyperplasia (*n* = 60)) than ACC and nonoriginal articles, were excluded. Articles assessing molecular biomarkers other than miRNAs were also removed. Full manuscripts were obtained as necessary to finalize eligibility (studies that were available only as abstracts were excluded). The articles were limited to those that presented information on the diagnostic, prognostic and therapeutic role of miRNAs in adrenocortical carcinoma. Twenty-nine studies qualified for inclusion in our study (Figure 1).

### 2.3. Data Collection Process

The full texts were carefully reviewed by the same reviewers who applied inclusion criteria. We collected the following data if available: the year of publication and name of the first author, the number of study participants, the pathologies, the study inclusion and exclusion criteria, the characteristics of the study participants, the type of samples that were used (tissue specimens or blood samples), the methods of miRNA isolation and the statistically significant test results.

## 3. Results

### 3.1. The Diagnostic Role of miRNAs

Twenty-two studies investigated the role of miRNAs in the diagnosis and prognosis of ACCs (Table 1). Fourteen studies [15,16,17,18,19,20,21,22,23,24,25,26,27,28] evaluated miRNAs’ expression in adrenal tissues frozen or paraben embedded, six studies in blood samples [29,30,31,32,33,34] and two in both tissue and blood samples of patients diagnosed with adrenocortical tumors [35,36].

MiR-483-5p is one of the most investigated miRNAs in ACCs, as it was found to be statistically significantly upregulated in ACCs compared with ACAs and/or healthy controls in 14 studies; 9 οf them [15,17,19,20,21,26,27,28,35] included adrenal tissue samples, whereas the remaining 4 studies included blood samples [29,30,32,34] and the last both tissue and blood samples [36]. Furthermore, overexpression of miR-210 was also reported in ACCs compared with ACAs and/or healthy controls in seven studies; six in tissue [16,19,20,21,27,28] and one in blood samples [29]. MiR-483-3p and miR-503 were also overexpressed in ACC tissue samples in six [19,20,21,22,27,28] and four [15,16,20,23] studies accordingly. MiR-483-3p was also found overexpressed in ACC plasma samples compared with ACAs in one study [36]. Furthermore, miR-184 was found upregulated in patients with ACC in five studies, four in tissue [16,20,27,36] and one in blood samples [29]. MiR-542-3p and miR-542-5p were also found upregulated in two studies in ACC tissues [15,27]. Finally, miR-139-5p and miR-181b were found upregulated in both ACC tissues and blood sample studies [15,20,29,35]. 

The most common downregulated miRNAs in ACCs compared with ACAs and/or normal adrenal tissue samples included the following: the downregulation of miR-195 was validated in eight studies, all at the tissue level [15,17,19,20,21,23,28], except one study that included both tissue and blood samples [35]. The downregulation of miR-335 was validated in six studies; six in tissue samples [15,18,20,23,28,35], and the last also included blood samples [35]. Finally, miR-497 and miR-214 were also found downregulated in tissue samples in five [19,20,23,28,35] and in three [16,20,28] studies, respectively.

It is noteworthy that, in two studies, miRNA expression was studied through microarray analysis solely, in eight studies through RT-PCR, and in eight studies, results were validated by both microarray and RT-PCR analyses. However, in two studies, there were differences between the results of microarray analysis and qRT-PCR. In particular, in the study of Soon et al., the expression of miR-7 was found to be significantly lower in ACCs compared with ACAs on qRT-PCR, a result that had not been found in microarray analysis [15]. Additionally, in the study of Chabre et al., while microarray analysis indicated that miR-483-5p expression was not significantly different between ACCs and ACAs, RT-qPCR analyses revealed marked upregulation of miR-483-5p in ACCs compared with the ACAs of the validation cohort [35].

Tissue- and blood-circulating miRNAs levels were not always concordant. In the study of Patterson et al. [17], miR-100 was found downregulated in ACC tissue samples, while in the study of Szabó et al. [29], miR-100 was upregulated in the blood of ACC patients compared to patients with ACAs. A lower expression of miR-34a in ACC tumor samples compared to ACA was also observed [23], while Patel et al. found a higher expression of circulating miR-34a in the blood of ACC patients compared to patients with ACA [30]. Moreover, in the study of Chabre et al., an inverse correlation between miRNA expression in tumor samples and circulating blood levels was observed [35]. In particular, although miR-376a levels were found upregulated in ACC tissue samples and mainly in patients with aggressive ACC, its levels were significantly decreased in blood samples of the same patients compared with controls or ACA patients. In the study of Decmann et al., differences in the expression between tissue and blood miRNA levels were observed [36].

Finally, in another study, circulating miR-483-5p levels were found significantly overexpressed in the blood of ACC compared to ACA patients, whereas no significant difference was observed in their urinary samples [34].

### 3.2. The Prognostic Role of miRNAs

Data from 10 studies, 7 in tissues [15,19,20,21,23,37,38], 2 in blood samples [31,39] and the last in both tissue and blood samples [35], investigating the role of miRNAs as prognostic biomarkers in patients with ACC, were analyzed (Table 2).

These miRNAs used as diagnostic markers to discriminate ACCs from ACAs seem to also differentiate aggressive from indolent ACCs. The upregulation of miR-483-5p in two studies in tissue [15,23] and in three in blood samples [31,35,39] has been associated with either short overall survival (OS), recurrence-free survival or disease progression. One study showed that high circulating levels of blood miR-483-5p postoperatively was associated with more than a four-fold increased risk of recurrence and was predictive of poor OS for ACC patients [39]. 

Moreover, overexpression of miR-503, miR-210 and miR-139-5p in the adrenal tissue of patients with ACC was associated with more aggressive behavior of the disease [19,21,23,35,38]. In particular, high miR-210 levels were associated with tissue necrosis and a high Ki-67 proliferation index [21]. The low expression of miR-195 was also observed in patients with aggressive ACCs in three studies (two in tissues, one in blood) [15,23,35]. MiR-139-5p and miR-376a levels were significantly upregulated in aggressive ACCs compared with non-aggressive ACC tumors samples, although no differences were observed in blood [35].

Eight out of ten studies [15,19,21,31,35,37,38,39] used Kaplan–Meier curves and the log-rank test to associate the expression of miRNAs with prognosis, while three of them [21,31,39] performed univariate and multivariate Cox proportional hazard analysis. One study [23] performed only Spearman correlation to analyze the correlation of miRNAs with distant metastases and disease progression. Finally, Assie et al. [20] did not select single miRNAs to assess their prognostic role in ACCs but identified miRNA clusters correlated with groups of ACC patients with different prognoses.

### 3.3. The Therapeutic Role of miRNAs

The therapeutic potential of miRNAs was tested in two studies (Table 3). In the first study [40], it was shown that miR-7 replacement in vivo inhibits ACC xenograft growth in models derived from both the adrenocortical cell line (H295R) and primary ACC cells. The second and more recent study attempted to identify differentially expressed miRNAs between ACC patients responsive to adjuvant therapy (mitotane, chemotherapy and radiotherapy) and ACC patients resistant to adjuvant therapy with progressive disease on adjuvant treatment. MiR-431 was the most downregulated miRNA in the resistant group when compared with the sensitive group. In vitro restoration of miR-431 enhanced the cytotoxic effects of doxorubicin and mitotane [41].

Several miRNAs were also validated as markers of treatment efficacy (Table 4). Circulating postoperative miR-105 blood levels were increased, whereas miR-483-5p decreased compared to preoperative levels in a small series of patients [35]. Postsurgical miR483 and miR-483-5p blood levels were also downregulated compared to presurgery levels in 27 ACC patients, although this decrease did not reach statistical significance [31]. Alteration in the expression of circulating miR-483-5p, miR-210, miR-181b and miR-184 levels was studied in four groups of patients with metastatic ACC: control, mitotane treated, 9-cis-retinoic acid treated and 9-cis-retinoic acid plus. It was found that only circulating blood miR-483-5p levels were significantly suppressed by the combined 9-cisRA + mitotane treatment in the ACC xenograft mouse model after treatment [42]. On the contrary, no significant changes were observed in the expression of tissue hsa-miR-483-5p between the four groups. In another preclinical study, levels of circulating blood miRNA-210 in the SW-13 tumor model were found to be elevated after combination therapy with etoposide, liposomal doxorubicin, liposomal cisplatin and mitotane, whereas no treatment-dependent changes were revealed for miR-483-5p [43].

### 3.4. The Oncogenic Role of miRNAs

The role of dysregulation of miRNAs in the oncogenic pathways of ACC has been studied in vivo and in vitro (Table 5, Figure 2). The miR-483 gene locus has been mapped to intron 2 of *IGF2* [44], one of the most commonly overexpressed genes in ACC [45]. The high expression of miR-483-5p and miR-483-3p observed in ACC could be correlated with the high expression of IGF2. Moreover, the p53 upregulated modulator of apoptosis (PUMA) expression was found significantly downregulated in ACCs and inversely correlated with miR-483-3p expression [44]. Additionally, it was demonstrated that miR-483-5p and miR-139-5p promoted ACC cell migration and invasion by suppressing the expression of two members of the N-myc downstream-regulated gene family *NDRG2* and *NDRG4* [38].

Dysregulation of four miRNAs was studied in vitro in human ACC cell lines [19]. Suppression of miR-483-5p and miR-483-3p expression led to a significant reduction in cell proliferation. Transfected cells with anti-miR-483-3p but not with anti-miR-483-5p resulted in a significant increase in apoptosis. Moreover, the overexpression of miR-195 or miR-497 resulted in a significant decrease in cell growth and induction of cell death through the suppression effect on both *TARBP2* and *DICER* genes [46]. Inhibition of *TARBP2* expression in human NCI-H295R ACC cells resulted in decreased cell proliferation and induction of cell apoptosis. Furthermore, the oncogenic mechanism of miR-497 could also be attributed to its ability to negatively regulate the expression of *MALAT1*, which, in turn, reversely competes for miR-497 binding to *EIF4E* [48]. It was observed that the overexpression of miR-497 and silencing of MALAT1 suppressed cellular proliferation and induced cell cycle arrest through downregulation of EIF4E expression. MiRNA-497 is a part of the miR-15 family cluster, located at the chromosomal region 17p13.1, in which there is a high frequency of loss of heterozygosity (LOH) in ACC compared to that of ACA neoplasms [50].

Additionally, miR-195 expression was inversely correlated with *ZNF367* expression [47]. *ZNF367* was overexpressed in ACCs compared to normal tissue and benign tumor and reduced cellular proliferation, invasion, migration and adhesion to extracellular proteins both in vitro and in vivo. Finally, molecular targets of miR-100 were also elucidated, such as IGF-1R and mammalian target of rapamycin (mTOR) signaling cascades [49].

Functional studies have demonstrated that miR-210 is a versatile molecule that regulates many aspects of hypoxia pathways, both in physiological and malignant conditions [51]. Although its role in ACC pathogenesis has not yet been elucidated, in the study of Duregon et al. [21], increased expression of miRNA-210 levels in tissue ACC samples was positively associated with necrosis and GLUT-1 expression. The inhibition of oxidative phosphorylation resulting from exposure to hypoxia leads to a stimulation of glucose transport, and this response is mediated by the enhanced function of glucose transporters, like GLUT-1 [52]. The relevance of miR-184, miR-503, miR-542-5p, miR-542-3p, miR-181b in the pathogenesis of ACC deserves further investigation. 

However, these miRNAs are involved in the regulation of proliferation, invasion, apoptosis and other processes in various tumor cells. For example, research has suggested that the miRNA-184 can play a role as a tumor suppressor by inhibiting the proliferation and invasion of glioma [53], oral cancer [54] and lung cancer cells [55], and it can act as an oncogene by inhibiting apoptosis of renal cancer cells [56]. MiR-503 inhibits the G1/S transition by downregulating cyclin D3 and E2F3 in hepatocellular carcinoma [57], it inhibits cell proliferation and invasion in glioma by targeting L1CAM [58], it targets PI3K p85 and IKK-β and suppresses the progression of non-small-cell lung cancer [58], and it inhibits cellular proliferation by targeting the AKT2 3′-UTR region in cervical cancer [59]. Moreover, there are many reports that demonstrate that miR-542-3p dysregulation is associated with several malignancies. For example, Rang et al. [60] reported that miR-542-3p can directly target the protooncogene PIM1 in melanoma, and its downregulation can enhance melanoma cell migration, invasion, and epithelial–mesenchymal transition (EMT) in vitro and in vivo. Yang et al. [61] demonstrated that miR-542-3p regulates cortactin (CTTN) in a targeted manner to modulate the growth and invasion of colorectal cancer cells. Althoff et al. [62] reported that miR-542-3p exerts its tumor-suppressive function in neuroblastoma, at least in part, by targeting survivin. Zhang et al. [63] also found that miR-542-3p downregulation induces cancer metastasis and hyperactivity of the TGF-β signaling pathway, thus promoting EMT and cancer progression in hepatocellular carcinoma. The role of the other mature sequence formed from pre-miR-542, miR-542-5p, has been described in tumors such as lung cancer [64], breast cancer [65], endometrial carcinosarcoma [66] and osteosarcoma [67]. Finally, the miR-181 family has been demonstrated to exert regulatory effects on tumorigenesis by modulating multiple signaling pathways, including PI3K/AKT, MAPK, TGF-b, Wnt, NF-κB and Notch pathways [68].

## 4. Discussion

A number of studies have reported the expression of miRNAs in ACCs. Earlier studies using microarray and RT-q-PCR techniques could only investigate known miRNAs, whereas later studies utilizing RNA sequencing could identify differentially expressed miRNAs, which had not been previously characterized. In the present review, analyzing these data across the included studies, we identified that miR-483-5p, miR-210, miR-483-3p, miR-184, miR-503, miR-542-3p, miR-542-5p, miR-139-5p and miR-181b were upregulated in ACC patients compared with ACA and/or healthy controls in multiple datasets either in tissue or blood samples. On the contrary miR-195, miR-335, miR-497 the miR-214 were downregulated in ACC patients compared with ACA and/or healthy controls either in tissue or in blood samples.

More than 50% of miRNA genes are located in cancer-associated genomic regions or in fragile sites, suggesting that miRNAs play an important role in the pathogenesis of cancer [69]. MiR-483-5p is one of the most investigated miRNAs in ACCs, both as a diagnostic and prognostic biomarker, and has been proven as the best single-gene malignancy marker [27]. In the study of Chabre et al. [35], miR-483-5p levels were undetectable in the blood of healthy controls, ACA and nonaggressive ACC patients, whereas high levels were detected in the serum of patients with aggressive ACC. In addition to circulating blood miR-483-5p, its urinary counterpart was evaluated in patients with adrenal tumors [34]. However, no significant difference was detected between ACC and ACA urinary samples. The lack of significance between ACC and adrenal myelolipoma in the expression of both tissue and plasma miR-483-5p and miR-483-3p might represent a limitation in the use of these markers, though [36].

The decrease in miR-483-5p blood levels after surgery in ACC patients suggests dynamic changes in serum miRNAs in response to surgical therapy [35]. This decrease was confirmed by another study [31] but did not reach statistical significance, probably due to the differences of sampling time in relation to the date of operation, as miRNAs deriving from the adrenal tumor before being removed may still be present in the bloodstream. Treatment-induced changes were also revealed for circulating miR-483-5p after systemic therapy in ACC patients [42]. 

Several miRNAs that seemed to be useful as differentiators between ACCs and ACAs are also promising prognostic indicators of ACCs. The statistically significant upregulation of miR-483-5p, miR-503, miR-210 and miR-139-5p and the downregulation of miR-19 were associated with poor clinical outcome in ACCs in most of the studies. Biomarkers that could predict the biological behavior of these tumors are essential in clinical practice, as they could identify high-recurrence-risk patients that need more intensive monitoring or adjuvant therapies and identify low-recurrence-risk patients that could avoid potential morbid therapies. Indeed, high miR-210 levels were found to be associated with ACC aggressiveness and poor prognosis, affecting the OS of these patients similarly with well-established prognostic factors such as mitotic count, Ki-67 proliferation index and increased expression of SF-1 [21]. Moreover, some miRNAs have been found differentially expressed in ACC histological variants. Prominent underexpression of miR-483-5p, miR-483-3p and miR-210 levels in adrenal tissues has been observed in oncocytic compared to the classical and myxoid histotype of ACC [21]. This interesting finding was interpreted through the prism of the positive correlation of the high levels of miRNA-210 expression with parameters of hypoxia, such as necrosis and GLUT-1, and aggressive biological behavior, such as mitotic rate and Ki-67 proliferation index, which are usually low in oncocytic tumors.

However, these results should be considered with great caution because the analysis of miRNAs expression, as well as its correlation with prognosis, differed among studies, either due to the different methodology used for molecular and/or statistical analysis. Several studies [15,19,35,37,38] used Kaplan–Meier curves and the log-rank test to associate miRNA (low vs. high) levels with worse prognosis. Only three studies [21,31,39] performed, in addition to the log-rank test, univariate and multivariate Cox proportional hazard regression analysis, including, however, different prognostic parameters in their multivariate model. In particular, Duregon et al. [21] included myxoid or classical ACC histotype (mitotic count ≥11, Ki-67 proliferation index ≥20) SF-1 protein expression and miR-210 and found that only mitotic count remains a significant prognostic factor. Salvianti et al. [31] included age, sex and miR483-5p and found that miR483-5p was associated with recurrence-free survival. Finally, Oreglia et al. [39] included tumor size, Ki67, ENSAT stage and miR-483-5p and found that miR483-5p was associated only with recurrence-free survival but not with OS. In addition, one study [31] performed only Spearman correlation to analyze the correlation of miRNAs with distant metastases and disease progression.

Another point of issue is the different cut-offs used for the expression of miRNA levels among the different studies. Receiver operating characteristic (ROC) analyses were performed to determine cut-off values in three studies [31,35,39], and only two studies [15,37] used the dichotomized relative to the median value to determine cut-off values. Three studies [19,21,35] did not mention the cut-off value they used, whereas Agosta et al. [38] used the same cut-off values with Chabre et al. [35] study. Moreover, there was heterogeneity in the compared groups included in the ROC analyses. In particular, Chabre et al. [35] compared ACC patients with aggressive tumors defined as recurring tumors or tumors that were already metastatic at diagnosis with patients with nonaggressive ACC tumors. Oreglia et al. [39] divided patients with ACC into two groups: patients who showed a recurrence within 3 years (group R < 3 years) and patients who showed no recurrence during the first 3 years of follow-up. Salvianti et al. [31] divided ACC patients based on low (stage 1/2) versus high (stage 3/4) disease stages. Furthermore, the studied population concerning ACC patients was heterogenous among studies. For example, Duregon et al. [21] included also other than the classical histological types of ACC (oncocytic and myxoid), whereas Oreglia et al. [39] performed analyses only on postsurgical blood samples of patients with ACC.

Finally, all studies used data of miRNA expression deriving from RT-PCR but one [15], which used data from microarrays analysis. In the study of Ozata et al. [19], only three out of six miRNAs were found to present a statistically significant prognostic role, and the microarray-based results were also validated by RT-PCR. 

Across several studies, differences in the expression between tissue and blood miRNA levels were observed, suggesting that the predictive role of blood miRNAs may be independent of tissue specimens. A potential explanation for this finding could be that released miRNAs do not reflect completely the cellular profile, as some miRNAs are retained or released selectively in the blood circulation [70].

Other components in the miRNA biogenesis pathway also seem to be useful as diagnostic and prognostic markers in adrenocortical tumors. Particularly, TARBP2, DICER and DROSHA miRNA-target genes are significantly overexpressed in ACCs when compared with adenomas and normal adrenal tissue samples [46]. A weak DICER1 protein expression is associated with reduced disease-free and OS serving as a predictor of recurrence in ACCs [71]. Furthermore, the top five upregulated target genes in ACCs, YWHAZ, GATA6, LDLR, BZW1 and IGFBP5, and five downregulated target genes, such as TXNIP, MAPKAPK5, PMAIP1, RAD51 and MICA, interact with several miRNAs [72].

Thus, identifying the relationships between miRNA signatures and ACCs could help better understand the underlying mechanisms and help develop new therapeutic strategies. Overexpression of miRNAs can be triggered by using synthetic miRNA mimics. Conversely, overexpressed miRNAs can be silenced by antagomiRs to restore miRNA balance in cancer networks [73]. For example, inhibition of miR-21 and miR-17-92 was associated with reduced tumor growth, invasion, angiogenesis and metastasis [74]. Indeed, the therapeutic potential of the miR-122 antagonist, miravirsen, in the treatment of hepatitis C was evident from a multicentric phase II trial [75]. Despite the great potential of miRNAs as novel therapeutic targets in the management of ACCs, there are a variety of technical challenges limiting the practical application of miRNA therapy in clinical practice, e.g., the availability of targeted delivery vesicles. Liposome delivery was the first delivery vehicle in clinical trials for miRNA [11]. Liposomal delivery of chemotherapeutics has already been studied in xenograft models of adrenocortical tumors. A significant reduction in tumor size was detected in an ACC xenograft model after a single treatment with anti-IGF1 receptor (IGF1-R) immunoliposomes (SSLD-1H7) [76]. Liposomally encapsulated miRNAs, in combination with cytostatic agents or alone, may represent a novel treatment option for ACC in the future.

## 5. Conclusions

Despite significant advances in the understanding of the molecular landscape of ACC, major efforts are still needed to improve diagnosis, surveillance and treatment of patients with ACC. MiRNAs detected both in adrenal tissue and in human body fluids can be envisaged as potential noninvasive biomarkers of malignancy and/or disease recurrence. Altering the expression of the miRNAs might eventually expand the rather limited therapeutic repertoire in the management of adrenal tumors. The role of miRNA regulation in ACC remains an area of active research with the potential to further enhance our understanding of its tumor biology and the molecular pathways involved.

## Figures and Tables

**Figure 1 biomedicines-09-01501-f001:**
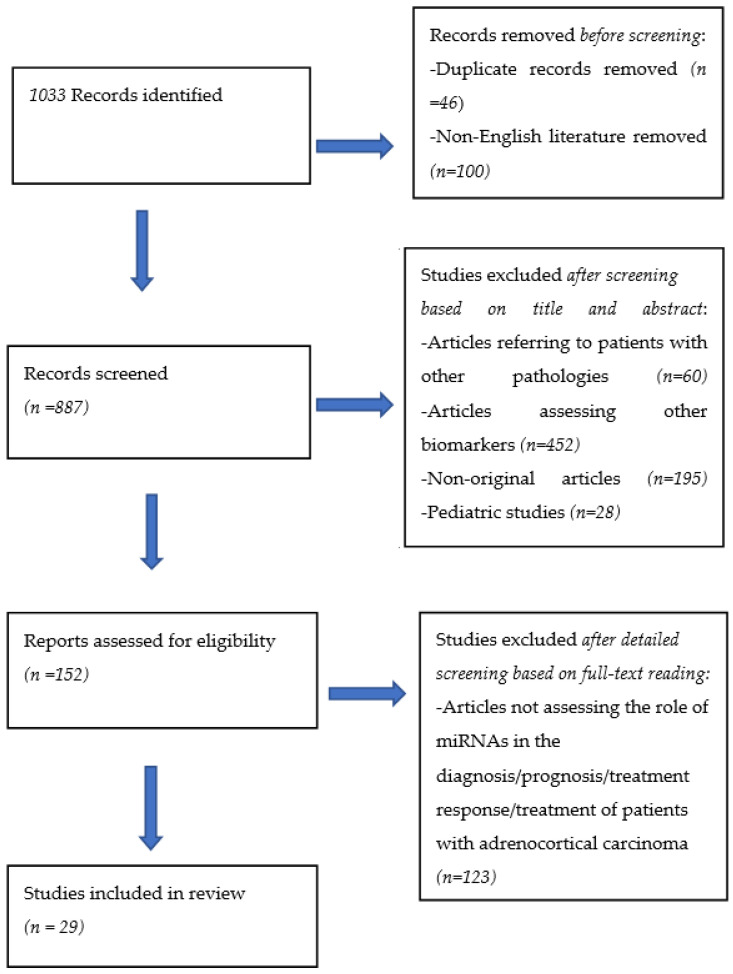
PRISMA flow diagram.

**Figure 2 biomedicines-09-01501-f002:**
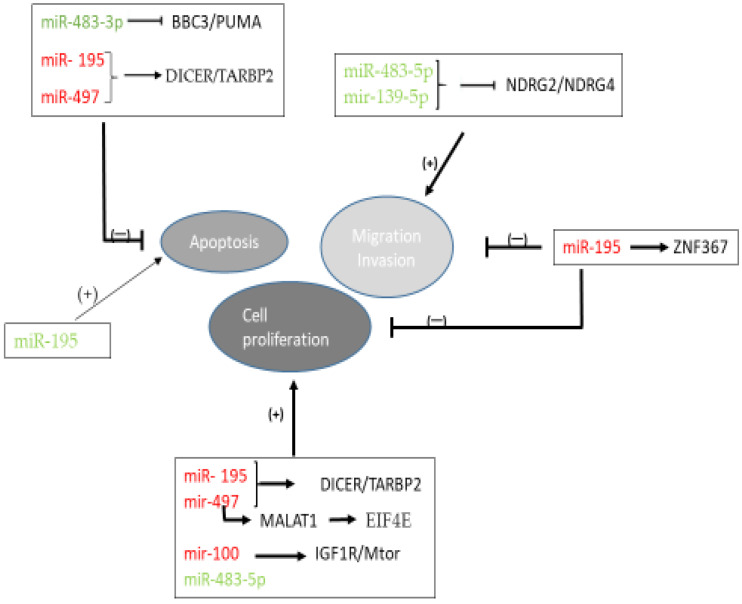
Schematic representation showing the miRNA- mediated mechanisms involved in ACC pathogenesis. Green fonts indicate upregulation and red fonts indicate downregulation of miRNAs. Arrows indicate stimulation, whereas T-arrows indicate inhibition. Abbreviations: BBC3; Bcl-2-binding component 3, EIF4E; eukaryotic translation initiation factor 4E, IGF1R; insulin-like growth factor 1 receptor; MALAT1; metastasis-associated lung adenocarcinoma transcript 1, miR: microRNA; mTOR: mammalian target of rapamycin; NDRG; N-myc downstream-regulated gene family member, PUMA; p53 upregulated modulator of apoptosis, TARBP2; transactivation response RNA-binding protein 2, ZNF367; zinc finger protein 367.

**Table 1 biomedicines-09-01501-t001:** Comparison of miRNAs’ expression in adrenocortical carcinomas (ACCs), adrenocortical adenomas (ACAs) and/or normal adrenal cortices (NACs).

Study	Cohort (n)	Method of miRNA Isolation	Sample	Dysregulated miRNAs				*p*-Value (for All Comparisons)
				ACC vs. ACA		ACC vs. NAC		
				Upregulation	Downregulation	Upregulation	Downregulation	
Soon et al., 2009 [15]	NAC(6), ACA(27), ACC(21)/ VC:ACA (9), ACC (10)	Microarray VC: RT-q-PCR	FFT	Microarray: miR-339-5p, miR-130b, **miR-483-5p**, miR-106b, miR-148b, miR-93, miR-135a, miR-320a, **miR-503**, miR-450a, **miR-542-3p**, miR-143, **miR-181b, miR-542-5p**	Microarray: **miR-335**, **miR-195**, miR-557, miR-708, miR-29c, miR-617, miR-647, let-7c, miR-202, -VC: **miR-195**, **miR-335**, miR-7 -Both: **miR-195**, **miR-335**	nd	VC: miR-7 (0.035)	<0.0001 (for comparisons with microarray analysis) <0.003 (for comparisons via RT-PCR)
Tombol et al., 2009 [16]	NAC(10) ACA(19), ACC(7)	TLDA	FFT	**miR-184**, **miR-210**, **miR-503**	**miR-214**, miR-511, miR-375	**miR-184**, **miR-503**	**miR-214**, miR-511, miR-375	<0.05
Patterson et al., 2010 [17]	NAC(21), ACA(26), ACC(10)/ VC:ACA(35), NAC(21), ACC(31)	Microarray VC: RT-PCR	FFT	Microarray: 5 miRNAs upregulated -VC: **miR-483-5p** -Both: **miR-483-5p**	Microarray: 18 miRNAs downregulated -VC: **miR-100**, miR-125b, **miR-195** -Both: **miR-100**, miR-125b, **miR-195**	nd	nd	<0.01 (for comparisons via microarray analysis) <0.05 (for comparisons via RT-PCR)
Ozata et al., 2011 [19]	NAC(4), ACA(26), ACC(22)/ VC: NAC(10), ACA(43), ACC(25)	Microarray VC: RT-qPCR	FFT	-Microarray: 55 miRNAs dysregulated -VC: **miR-483-3**, **miR-483-5**, **miR-210**, miR-21 -Both: **miR-483-3p**, **miR-483-5p**, **miR-210**, miR-21	Microarray: 55 miRNAs dysregulated -VC: miR-1974, **miR-195**, **miR-497** -Both: miR-1974, **miR-195**, **miR-497**	Microarray: 42 miRNAs dysregulated -VC: **miR-483-3**, **miR-483-5**, **miR-210**, miR-21 -Both: **miR-483-3**, **miR-483-5**, **miR-210**, miR-21	Microarray: 42 miRNAs dysregulated -VC: miR-1974 ( **miR-195**, **miR-497**, -Both: miR-1974, **miR-195**, **miR-497**	<0.05 (for comparisons via microarray analysis) <0.03 (for comparisons via RT-PCR)
Schmitz et al., 2011 [18]	NAC(4), ACA(9), ACC(7)/ VC: ACT(15)	Microarray VC: RT-qPCR	FFPE	Microarray: 89 miRNAs upregulated (vs. Conn syndrome), 35 miRNAs upregulated (vs. Cushing syndrome)	Microarray: 38 miRNAs downregulated (vs. Conn syndrome), 159 miRNAs downregulated (vs. Cushing syndrome) -VC: **miR-335**, miR-675, miR-139-3p -Both: **miR-335**, miR-675, miR-139-3p	Microarray: 62 miRNAs upregulated	Microarray:74 miRNAs downregulated -VC: miR-139-3p, **miR-335**, miR-675 -Both: **miR-335**, miR-675, miR-139-3p	<0.05 (for comparisons via microarray analysis) <0.001 (for comparisons via RT-PCR)
Chabre et al., 2013 [35]	ACA(6), ACC(12)/ VC: NAC(3), ACA(10), ACC(18)	Microarray VC: RT-qPCR	FFT	Microarray: **miR-503**, miR-514, miR-509-3p, miR-93, miR-148B, miR-508-3p, miR-513A-5p -VC: **miR-483-5p**	Microarray: **miR-335**, **miR-195**, **miR-497**, miR-199a-3p, miR-199a-5p -VC: **miR-335**, **miR-195** -Both: **miR-335**, **miR-195**	nd	nd	<0.05
Chabre et al., 2013 [35]	NAC(19), ACA(14), ACC(23)	RT-qPCR	Serum	miR-139-5p	**miR-195**, **miR-335**, **miR-376a**	nd	**miR-195**, **miR-335**, **miR-376a**	<0.05
Patel et al., 2013 [30]	ACA(22), ACC(17)	RT-qPCR	Serum	**miR-34a** **miR-483-5p**	-	nd	nd	<0.011
Szabo et al., 2013 [29]	ACA (12), ACC (13)/ VC: ACA(4), ACC(4)	Microarray VC: RT-qPCR	Serum	VC: **miR-100**, **miR-181b**, **miR-184**, **miR-210**, **miR-483-5p**	Microarray: miR-192, miR-197	nd	nd	<0.05
Assie et al., 2014 [20]	NAC(3), ACC(45)	RNA sequencing	FFT	Nd	Nd	**miR-483-3**, **miR-483-5p**, **miR-210**, **miR-503**, **miR-184**, **miR-139-5P**, **miR-376a**	**miR-195**, **miR-335**, **miR-214**, **miR-497**	Nd
Duregon et al., 2014 [21]	ACA(47), ACC(51)	RT-qPCR	FFT	**miR-483-3p**, **miR-483-5p**, **miR-210**	**miR-195**	nd	nd	<0.0001
Wang et al., 2014 [22]	ACA(25), ACC(25)	In situ hybridization	FFPE	**miR-483-3p**	-	nd	nd	<0.001
Feinmesser et al., 2015 [23]	ACA(25) ACC(8)/ VC: ACA(4), ACC(11)	Microarray VC: RT-qPCR	FFPE	Microarray: Over a dozen miRNAs dysregulated -VC: **miR-503** -Both: **miR-503**	Microarray: Over a dozen miRNAs dysregulated -VC: **miR-34a**, and **miR-497** (combination) **miR-335**, **miR-195** -Both: **miR-34a** and **miR-497** (combination) **miR-335** **miR-195**	nd	nd	<0.05
Gara et al., 2015 [24]	NAC(21), ACA(26), ACC(10)	Microarray	FFT	miR-9, miR-25, miR-124, miR-183, miR-185, miR-206	-	miR-9, miR-25, miR-124, miR-183, miR-185, miR-206	-	<0.05
Wu et al., 2015 [25]	ACA(21) ACC(11)	RT-qPCR	Tissue	-	miR-205	nd	nd	0.008
Zheng et al., 2016 [26]	NAC(120) ACC(79)	RNA sequencing	FFT	Nd	Nd	miR-10-5p, **miR-483-5**, miR-22-3p, miR-508-3p, miR-509-5p, miR-340, miR-146a, miR-21-3p, miR-21-5p, miR-509-3p,	-	<0.05
Koperski et al., 2017 [27]	NAC(8) ACA(8) ACC(7)/ VC: NAC(10), ACA(10), ACC(8)	RNA sequencing VC: RT-PCR	FFPE	RNA sequencing: miR-503-5p, miR-450a-5p, **miR-542-5p**, **miR-483-3p**, **miR-542-3p**, miR-450b-5p, **miR-210**, **miR-483-5p**, miR-421, miR-424-3p, miR-424-5p, miR-598, miR-148b-3p, **miR-184** miR-128 -VC: **miR-483-3p** -Both: **miR-483-3p**	nd-	RNA sequencing: miR-503-5p, miR-450a-5p, miR-542-5p, **miR-483-3p**, miR-542-3p, miR-450b-5p, **miR-210**, **miR-483-5p**, miR-421, miR-424-3p, miR-424-5p, miR-598, miR-148b-3p, **miR-184**, miR-128	-	<0.05
Koduru et al., 2017 [28]	ACA(30) ACC(45)	RNA sequencing	Tissue	**miR-483-3p**, **miR-483-5p**, miR-153, miR-135, miR-514, **miR-210**	**miR-497**, **miR-195**, **miR-335**, **miR-214**, miR-199	nd	nd	<9*10^−6^
Perge et al., 2017 [32]	ACA(6), ACC(6)/ VC: ACA(18), ACC(16)	Microarray VC: RT-PCR	Plasma	Microarray: miR-101, **miR-483-5p** -VC: miR-101, **miR-483-5p** -Both: miR-101, **miR-483-5p**	-	nd	nd	<0.05 (for comparisons via microarray analysis) <0.0052 (for comparisons via RT-PCR)
Salvianti et al., 2017 [31]	ACA(13), Stage ¾ ACC(27), NAC(10)	RT-PCR	Serum	miR-483	-	miR-483	-	<0.018
Decmann et al., 2018 [36]	ACA(10), ACC(10)/ VC: ACA(14), ACC(12)	Next-generation sequencing VC: RT-PCR	FFPE	Microarray: **miR-184** **miR-483-5p** **miR-483-3p** miR-183-5p -VC: **miR-184**, **miR-483-5p**, miR-183-5p -Both: **miR-184**, **miR-483-5p**, miR-183-5p	-	-	-	<0.001 (for comparisons via microarray analysis) <0.01 (for comparisons via RT-PCR)
Decmann et al., 2018 [36]	ACA(11), ACC(11)	RT-PCR	Plasma	**miR-483-5p**, **miR-483-3p**	-	-	-	<0.05
Perge et al., 2018 [33]	ACA(26): NFA(13), CPA(13), ACC(9)	RT-PCR	Plasma	miR-22-3p (related to NFA,) miR-27a-3p (related to NFA), miR-320b (related to CPA and NFA) miR-210-3p (related to NFA)	-	nd	nd	<0.05
Decmann et al., 2019 [34]	ACA(23) ACC(23)	RT-PCR	Serum	**miR-483-5p**	-	Nd	nd	<0.0001

miRNAs found dysregulated in more than one study are highlighted in bold. Abbreviations: ACC, adrenocortical carcinomas; ACA, adrenocortical adenomas; NAC, normal adrenal cortices; ACT, adrenocortical tumors; NFA, nonfunctioning adenomas; CPA, cortisol-producing adenomas; TLDA, TaqMan low-density array; VC, validation cohort; FFT, fresh frozen tissue; FFPE: formalin-fixed paraffin-embedded; nd, no data.

**Table 2 biomedicines-09-01501-t002:** miRNAs as prognostic biomarkers in ACCs.

Study	Cohort (n)	Methodology	Sample	miRNAs	Outcome Studied	*p*-Value
Soon et al., 2009 [15]	18	Microarray	Tissue	↑ **miR-483-5p**, ↓ **miR-195**	OS (log-rank test)	0.036 0.035
Ozata et al., 2011 [19]	22	qRT-PCR	Tissue	↑ **miR-503**, ↑ miR-1202, ↑ miR-1275	OS (log-rank test)	0.006 0.005 0.042
Chabre et al., 2013 [35]	21	qRT-PCR	Tissue	↑ **miR-139-5p**, ↑ miR-376a, ↑ miR-376b, ↑ miR-376c	Local and distant recurrences	<0.0001 <0.0001 <0.05 <0.05
Chabre et al., 2013 [35]	21	qRT-PCR	Serum	↑ **miR-483-5p**, ↓ **miR-195**	RFS and OS (log-rank test)	0.0004/0.0005 0.0014/0.0086
Assie et al., 2014 [20]	45	RNA sequencing	Tissue	↑ Mi3 miRNA cluster ↓ miR-508-3p, ↓ miR-509-3p, ↓ miR-513-3p, ↓ miR-514	OS	nd
Duregon et al., 2014 [21]	51	qRT- PCR	Tissue	↑ **miR-210**	OS (log-rank test/multivariate Cox model *)	0.046/0.2195
Faria et al., 2014 [37]	28	qRT-PCR	Tissue	↑ miR-9	RFS and OS (Log-rank test)	0.01/0.012
Feinmessser et al., 2015 [23]	17	qRT-PCR	Tissue	↑ miR-483-3p, ↑ **miR-483-5p**, ↑ miR-10b, ↑ miR-513-5p, ↑ miR-487a ↑ **miR-503**, ↑ **miR-210**, ↓ miR-497, ↓ miR-34a, ↓ miR-214, ↓ miR-99a, ↓ miR-125b, ↓ **miR-195**, ↓ miR-30c, ↓ miR-15a, ↓ miR-335, ↓ miR-345, ↓ miR-708, ↓ miR-29c	Distant metastases and disease progression (Spearman correlation)	<0.05
Salvianti et al., 2017 [31]	21	qRT-PCR	Serum	↑ **miR-483-5p**	RFS and OS (Log-rank test/multivariate Cox model **)	RFS: 0.027/0.026 OS: 0.001/ns
Agosta et al., 2018 [38]	20	qRT-PCR	Tissue	↑ **miR-139-5p**	OS (log-rank test)	<0.0001
Oreglia et al., 2020 [39]	26	qRT-PCR	Serum	↑ **miR-483-5p**	RFS and OS (log-rank test/multivariate Cox model ^≠^)	RFS: 0.0005/0.011 OS: 0.007/0.150

Upwards arrows (**↑)** indicate upregulation and downwards arrows (↓) indicate downregulation of miRNAs. miRNAs found dysregulated in more than one study are highlighted in bold. * Multivariate model: myxoid or classical histotype, mitotic count ≥ 11, Ki-67 proliferation index ≥ 20, SF-1 protein expression and miR-210. ** Multivariate model: age, sex and miR483-5p. ^≠^ Multivariate model: tumor size, Ki67, ENSAT stage and miR-483-5p. Abbreviations: aACC, aggressive adrenocortical carcinoma; naACC, nonaggressive adrenocortical carcinoma; nd, no data; OS, overall survival; RFS, recurrence-free survival; qRT-PCR, quantitative real-time -PCR.

**Table 3 biomedicines-09-01501-t003:** MiRNA-based ACC treatments.

Study	Pathology	miRNA-Based Treatment	Functional Role of miRNAs in ACC	Type of Study	Outcome
Glover et al., 2015 [40]	Metastatic ACC	Replacement therapy: 10 doses of miR-7	Cell proliferation reduction by G1 cell cycle arrest induction	In vivo: Patient-derived xenograft in mice	Tumor reduction
Kwok et al., 2019 [41]	Metastatic ACC	Replacement therapy: miR-431	Cell death, EMT reversal	In vitro: ACC H295R cells and primary-derived ACC cells	Increased ACC cell response to doxorubicin and mitotane

Abbreviations: EMT, epithelial–mesenchymal transition; ACC, adrenocortical carcinoma.

**Table 4 biomedicines-09-01501-t004:** miRNAs as therapeutic biomarkers for ACC.

Study	Type of Study	Treatment	Post-Treatment Marker’s Level	Source	*p*-Value
Chabre et al., 2013 [35]	Clinical	Surgical removal	↓ miR-483-5p, ↑ miR-195	Serum	<0.05
Nagy et al., 2015 [42]	In vivo: Patient-derived xenograft in mice	Combined 9-cis retinoic acid + mitotane treatment.	↓ miR-483-5p	Serum	0.028
Jung et al., 2016 [43]	In vivo: SW-13 xenograft	LEDP-M treatment	↓ miR-210	Serum	<0.05

Upwards arrows (**↑)** indicate upregulation and downwards arrows (↓) indicate downregulation of miRNAs. Abbreviations: LEDP-M, etoposide, liposomal doxorubicin, liposomal cisplatin, mitotane.

**Table 5 biomedicines-09-01501-t005:** miRNAs most frequently dysregulated in ACC compared to NAC or ACA.

miRNA	Expression	Role of miRNAs in ACC Pathogenesis
**miR-483-5p**	Upregulated	Promotes cell proliferation [19] Promotes cell migration and invasion by suppressing expression of NDRG2 ^b^ [38]
**miR-195**	Downregulated	Overexpression of miR-195 leads to decrease in cell growth and induction of cell death [19] Inhibits TARBP2 ^b^ and DICER gene expression [46] Targets ZNF367 ^c^ and regulates cellular invasion [47]
**miR-210**	Upregulated	Related to hypoxia parameters [21]
**miR-483-3p**	Upregulated	Represses the proapoptotic gene BBC3/PUMA ^d^ [44] MiR-483-3p silencing suppresses cell proliferation and induces apoptosis [19]
**miR-335**	Downregulated	Potential target genes are: NEBL ^e^, C8orf44 ^f^, SEC14L5 ^g^, PRDM2 ^h^, PLEKHK1 ^i^, KPNA6 ^j^, TNFAIP ^k^, ONECUT2 ^l^, UNC5D ^m^, MMAA ^n^ [18]
**miR-497**	Downregulated	Overexpression of miR-497 reduces cell growth and induces apoptosis [19] Inhibits DICER and TARBP2 ^b^ gene expression [46] Represses the lncRNA MALAT1 ^o^ and targets the EIF4E ^p^. Its overexpression suppresses cellular proliferation and induces cell cycle arrest [48]
**miR-184**	Upregulated	Unknown
**miR-503**	Upregulated	Unknown
**miR-214**	Downregulated	Unknown
**miR-139-5p**	Upregulated	Promotes cell migration and invasion by suppressing expression of NDRG4 ^a^ [38]
**miR-100**	Up- or downregulated	Regulates the IGF-mTOR ^q^-raptor signaling pathway at multiple levels [49]
**miR-542-3p**	Upregulated	Unknown
**miR-542-5p**	Upregulated	Unknown
**miR-181b**	Upregulated	Unknown

^a^ NDRG2, NDRG4: N-myc downstream-regulated gene family member; ^b^ TARBP2: transactivation response RNA-binding protein 2; ^c^ ZNF367: zinc finger protein 367; ^d^ PUMA:p53 upregulated modulator of apoptosis; ^e^ NEBL: nebulette (actin-binding Z-disk protein); ^f^ C8orf44: putative uncharacterized protein C8orf44; ^g^ SEC14L5: SEC14-like 5; ^h^ PRDM2: PR domain zinc finger protein 2; ^i^ PLEKHK1: pleckstrin homology domain containing; ^j^ KPNA6: importin subunit a-7 (karyopherin subunit a-7); ^k^ TNFAIP1: BTB/POZ domain-containing protein TNFAIP1 (tumor necrosis factor); ^l^ ONECUT2: one-cut domain family member 2; ^m^ UNC5D: Unc-5 homolog D; ^n^ MMAA: methylmalonic aciduria type A protein; ^o^ MALAT1: metastasis-associated lung adenocarcinoma transcript 1; ^p^ EIF4E: eukaryotic translation initiation factor 4E; ^q^ mTOR: mammalian target of rapamycin.

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
