# Peer review of "The Diagnostic, Prognostic and Therapeutic Role of miRNAs in Adrenocortical Carcinoma: A Systematic Review"

_biomedicines, 2021, doi:10.3390/biomedicines9111501_

Round 1

Reviewer 1 Report

A systematic review on miRNAs in adrenocortical carcinoma which I evaluated is the fourth one on this topic in the last three years. Some data presented in this manuscript were discussed in these previously published review articles.

For example, the paper referenced as #11 (Hassan et al) is a review article published in 2017. It shows and explains results from 11 studies showing data related to mi-RNA/ACC. The same set of data is presented in the review paper „Key MicroRNA’s and Their Targetome in Adrenocortical Cancer“, published in Cancers last year and not cited here.

All these data are again presented in this manuscript. I see no reason for it.

In addition to obvious data duplication, there are some incorrect statements.

Wang et al (ref #21) did not perform microarray analysis.

Ref #24 – only RT-qPCR was performed, and not microarray.

A small part of the text was rewritten from the review written by Mizrak et al. „The Role of Biomarkers in Adrenocortical Carcinoma: A Review of Current Evidence and Future Perspectives“, Biomedicines, 2021.

Some differences with respect to histology of ACC and specific micro-RNA expression were observed in some studies (Duregon et al, 2014). These are the data that should be presented and discussed.

In Table 5, the authors should be focused on experimentally confirmed targets (transcription from the 2nd intron of IGF2 should not be presented under „Targets in the adrenal gland“). Even if the mRNA target of a specific micro-RNA in ACC is not experimentally confirmed, the authors should discuss crucial targets discovered in some other types of malignant tumors. For example, micro-210 was shown to be closely related to regulation of hypoxia related genes.

Based on these objections, I suggested to the authors that do NOT present and discuss data that have been already published in recent review papers. This suggestion of mine primarily relates to differential expression of micro-RNAs in ACC vs. normal vs. ACA.

Instead, I suggest them to transfer their focus on clinical significance of differentially expressed micro-RNAs as prognostic and predictive biomarkers in ACC and change the title of the manuscript accordingly.

Thank you.

Reviewer 2 Report

  1. This review is acceptable. The authors provide a systemic review for the distribution and biomarker potentials of miRNA in adrenocortical carcinoma. Table 1 summarized miRNA expression levels in adrenocortical carcinoma, benign tumor or normal adrenal cortices. And in adrenocortical carcinoma, Table 2 summarized prognostic miRNA biomarkers, Table 3 showed some miRNA-based treatment, Table 4 summarized therapeutic miRNA biomarkers, and Table 5 summarized most frequently dysregulated miRNA. It should be informative and useful for readers of this field.
  2. There is no any figures to help readers to quickly realize the miRNA regulation in adrenocortical carcinoma.
  3. A few typographic mistakes might be revised before submission.

Round 2

Reviewer 1 Report

The revised version of the manuscript „MiRNAs in adrenocortical carcinoma. A systematic review” is still lacking some important data. If we are to be focused on “prognostic” aspect of the ACC, then please extract data as precise as possible. Make a difference between microarray and RT-qPCR (of note, Soon, Ozata and Chabre DID perform the microarray first) and only validation was made by real-time PCR. Stating the methods used IS important because the potential limitation of the methods may be relevant for some data that was not reproducible among studies. For example, Duregon performed a targeted research, exploring only five micro-RNAs by RT-qPCR. Among them was hsa-miR-483-5p.Still, contrary to some other studies, here, in Duregon’s study, it was not shown as a marker of shorter OS. Why is that so? Where there any differences related to univariate vs. multivariate analyses? Please, consider how the authors statistically test their hypotheses. The statistical analyses applied in some references cited and presented in Table 2 significantly differ.

Again, the role of any micro-RNA has little to do with its chromosomal location (unless the locus is lost). I specifically stated that THE ROLE of miR-483-5p primarily depends on its target transcript, and not the primary (imprinted or LOI) IGF2 transcript, where it originates.

If redundancy of data presented already is significant – and it is, then, please, offer to the readers at least one new aspect associated with studies cited in a precise and knowledgeable fashion. If these facts are not considered and carefully put in place, this review does not offer anything new.

Thank you.

Round 3

Reviewer 1 Report

The authors have properly addressed my concerns and suggestions. This review now have some interesting new elements.

Thank you.